# Trends in IT and Life Sciences Education based on AI Tools

## Abstract

AI-based tools such as ChatGPT are currently reshaping the landscape of higher education and significantly impacting student learning and academic achievement in multiple fields including mathematics, biophysics, bioinformatics. Gamification is the use of game mechanics to promote students' engagement to problem-solving in non-game situations. Gamification has been widely used for training following education in IT and AI. We review here trends in AI education based LLM (Large Language Models) and interactive training models and online courses.

## 1 Introduction

Artificial intelligence (AI) models, like Chat Generative Pre-Trained Transformer (OpenAI, San Francisco, CA), have recently gained significant popularity due to their ability to make autonomous decisions and engage in complex interactions (Hallquist et al., 2025). Here we review trends in education using LLM (Large Language Models) and interactive models based on the recent review and new tools focusing on Life sciences (Orlova, Orlov, 2025; Basha et al., 2025).

The mathematical competencies, defined as the ability to model, interpret, and solve problems through logical reasoning, are reinforced when integrated with critical thinking skills such as evaluation, argumentation, and evidence-based decision-making (Alvarez-Tinajero et al., 2026). The challenges include new methodological development using AI tools, technical base such as web-based and online training courses.

Applications for education courses for undergraduate students have shown importance of module and interactive courses. Course-based undergraduate research experiences represent a transformative pedagogical approach. Regarding software environment used internationally we note main programming tools and environments - R, Python, MATLAB (R Core Team 2021; Van Rossum and Drake, 2009; The Math Works, 2020). Interactive digital notebooks (Jupyter notebooks) have been used in an educational setting for bioinformatics teaching. For digital training courses, it is possible to reuse, repurpose existing educational materials, sample data and programs. To keep the standards, a set of rules have been developed based on FAIR (Findable, Accessible, Interoperable and Reusable) principles (Bacon et al., 2022).

Research on technology-enhanced higher education (TEHE) has been active and influential in educational technology (Chen et al., 2024). The analysis of topics highlighted research hotspots and emerging themes such as Massive Online Open Courses, AI and big data in education, Gamification and engagement, Learning effectiveness and strategies. We also note current transfer from classical textbook on bioinformatics to interactive courses and interactive textbooks, Wikipedia-style "Edit" links, to online accessible resources for bioinformatics learning (Orlova, Orlov, 2025).

Active learning assumes group problem-solving, the use of personal response systems. Gamification is the use of game mechanics to promote students' engagement to problem-solving in non-game situations (Freeman et al., 2024). Gamification has been widely used for training following education in computer sciences, medicine and AI (Aloum et al., 2025; Fadous et al., 2025; Chiuchiolo et al., 2026; Alvarez-Tinajero et al., 2026).

In general, bioinformatics education in the form of short courses, distant-learning, self-learning programs, and now generative models had passed its way to modern development of AI tools. Besides

the research problem, AI has diverse effects on education, both positive and negative (Shool et al., 2025). In particular, the problems of new educational approaches in bioinformatics have been discussed at the students' workshops organized at First Sechenov Moscow State Medical University of the Russian Ministry of Health in Moscow. The Digital Chair of Sechenov University presents educational materials in Russian extending universities' initiatives on IT education and digital transformation. As an example of the courses output, the training on computer gene network reconstruction and annotation have been effective for education of medical students leading to series of the students' project publications.

The perspectives on how to integrate tools like ChatGPT into bioinformatics have been discussed recently (Orlova, Orlov, 2025). Generative artificial intelligence has delivered promising results in drug discovery and development. Integration of LLM into learning and teaching practice improves digital competitiveness for medical and health science students. Finally, we highlight the latest problems related to ethical concerns on AI applications in education.

## 2 APPLICATIONS OF AI FOR MEDICAL EDUCATION

Series of AI applications for medical decision support naturally applied for education (Hallquist et al., 2025; Shool et al., 2025; Verghese et al., 2025).

Large language models (LLMs) and agent systems are increasingly transforming scientific discovery, driving progress across chemistry, biology, materials science, and physics. Large language models perform well on general medical benchmarks, but their ability to reason about rare diseases (RDs) remains unclear. Rather than challenge LLMs to diagnose a limited number of cases that are unlikely to represent all RDs or RD-associated genes, we instead sought to comprehensively probe LLM understanding of RD-associated genes and phenotypes. Groza et al. (2026) systematically evaluated six leading general-domain LLMs (GPT-4, Claude 3.7, Llama-3.3 70B, Gemma-2 27B, Llama-3.2, and Phi-4) for their ability to generate core phenotypic features and causal genes required to support reasoning for about 10,000 Orphanet diseases. Outputs were mapped to Human Phenotype Ontology (HPO) terms and HGNC gene symbols and compared with curated references using set overlap, semantic similarity, and disease ranking via the likelihood ratio interpretation of clinical abnormality (LIRICAL) framework applied to 8,000 patient Phenopackets. Commercial models, particularly GPT-4 and Claude, achieved over 60 percents recall for gene associations but struggled with precise phenotype recovery. Despite low exact overlaps, moderate semantic similarity scores indicated partial alignment with curated data. When used in LIRICAL, LLM-derived phenotypic profiles yielded ranking performance close to that of gold standard profiles, although direct diagnostic accuracy remained limited. Interestingly, convergent non-curated terms across models suggest potential for hypothesis generation. Current generalist LLMs lack the precision to replace curated RD knowledge bases but offer complementary, semantically relevant information. The results support hybrid approaches that combine expert curation with selectively integrated LLM outputs to enhance and scale ontology-driven RD diagnostics (Groza et al., 2026).

Recent comprehensive review of the literature was conducted across PubMed, Scopus, Web of Science, IEEE Xplore, and arXiv databases, encompassing both peer-reviewed and preprint studies using 761 studies revealed growing interest in leveraging LLM tools in clinical settings (Shool et al., 2025). Studies were screened against predefined inclusion and exclusion criteria to identify original research evaluating LLM performance in medical contexts. While general-domain LLMs, particularly ChatGPT and GPT-4, dominated evaluations (93.5 percents), medical-domain LLMs accounted for only 6.5 percents.

## 3 BIOINFORMATICS COURSES AND EDUCATIONAL SUPPORT

Bioinformatics education was defined as priority area for Bioinformatics Grand Challenges Consortium (Işık et al., 2023; Khan et al., 2024). Other discussed challenges and priority areas include data visualization, statistical genetics, single-cell data science, biological engineering, evolutionary and population genetics, and dynamic modeling (Rastogi, 2023).

Applications for education courses for undergraduate students had shown importance of module courses (Dill-McFarland et al., 2021). The postsecondary training in bioinformatics, delays behind

Table 1: Trends and publication on AI in medical education

| CHALLENGE | REFERENCE |
| --- | --- |
| Active learning | Chiuchiolo et al., 2026 |
| Gamification | Alvarez-Tinajero et al., 2026 |
| Serious games | Fadous et al., 2025 |
| Web-tools | Ortiz Martín et al., 2025 |
| Self-assesment | Aloum et al., 2025 |
| Knowledge extraction | Chiuchiolo et al., 2026 |

current demand. Participating in scientific research is essential for undergraduate students that major in natural sciences, public health, biomedicine to develop their skills in IT (Bennett and Page, 2022). Course-based undergraduate research experiences represent a transformative pedagogical approach (Zong et al., 2025).

Note the pandemic and isolation challenges related to the distant form of bioinformatics courses organization and distant learning. In 2020 when the coronavirus pandemic hit, the training activities were transformed. The pandemic further emphasized the need for increased online education (Ras et al., 2021).

Regarding software environment used we note main programming tools and environments - R, Python, MATLAB (R Core Team 2021; Van Rossum and Drake, 2009; The Math Works, 2020). Interactive digital notebooks (Jupyter notebooks) have been used in an educational setting for bioinformatics teaching (Davies et al., 2020). For digital training courses, it is possible to reuse, repurpose existing educational materials, sample data and programs. To keep the standards a set of rules have been developed based on FAIR (Findable, Accessible, Interoperable and Reusable) principles (Garcia et al., 2020; Wijnbergen et al., 2025).

Education in computer science, developing computer skills, programming and general informatics are necessary backgrounds for the students and post-graduates. We note current transfer from classical textbook on bioinformatics to interactive courses and interactive textbooks (Carvalho-Silva et al., 2018), Wikipedia-style "Edit" links (Kar, 2021). Online accessible resources for bioinformatics learning are available, for example Rosalind (Searls, 2014). Classical textbooks and courses are focused on theory, as well as on specific programming skills such as Python or the Unix shell (Wilson, 2016).

Community-based development of the training courses is a new trend in sharing the resources. It could be presented by Network for the Integration of Bioinformatics into Life Science Education (Ryder et al., 2020).

## 4 ACTIVE LEARNING AND GAMIFICATION

New learning methodologies in bioinformatics such as active learning have been recently discussed (Ortiz Martín et al., 2025; Aloum et al., 2025). Active learning assumes group problem-solving, the use of personal response systems. Gamification offers a powerful tool to foster engagement and improve the educational journey of medical professionals. By integrating interactive and motivational elements into training, the gamification not only boosts knowledge acquisition, but also addresses the monotony of traditional methods, encouraging deeper cognitive engagement and reducing errors in practice (Fadous et al., 2025).

The implementation of an active learning approach for undergraduate students in life sciences was developed at the University of Malaga in Spain (Ortiz Martín et al., 2025). The studies indicated that mathematical competencies, defined as the ability to model, interpret, and solve problems through logical reasoning, are reinforced when integrated with critical thinking skills such as evaluation, argumentation, and evidence-based decision-making (Alvarez-Tinajero et al., 2026).

Gamification is the use of game mechanics to promote students' engagement to problem-solving in non-game situations. Gamification has been widely used for training in bioinformatics, mathe-

matics, medical education, self-assessments (Waruingi et al., 2023; Alvarez-Tinajero et al., 2026; Chiuchiolo et al., 2026). Bioinformatics courses follow this trend (Fry, 2024; Oestreich and Guy, 2022). Mello and co-authors used QR codes (quick response) in the context of environmental DNA studies codes for students to mark DNA (Mello et al. 2017).

A multiplayer educational game is hosted on a web-based platform, enabling students to engage in critical thinking exercises remotely (Fadous et al., 2025). The implementation of 3 open-access, web-based pharmacology games tailored for medical students showed better learning indicators for the gamers (Aloum et al., 2025). The forms for students' involvement might be diverse, such as board games, online tools, home tasks and quests. An analogy to the Pokémon GO game of the Pokémon franchise was suggested to engage students in bioinformatics (Nunes et al., 2021).

## 5 EDUCATIONAL COURSES IN RUSSIA

Several research centers in Russia work on development of education courses on bioinformatics and AI (Nawaz e al., 2024; Orlova and Orlov, 2025). In addition to the School for young scientists in Novosibirsk, Russia (Orlov et al. 2023), the problems of new educational approaches in bioinformatics have been discussed at the students' workshops organized at First Sechenov Moscow State Medical University of the Russian Ministry of Health in Moscow (Turkina et al., 2023). The website of Digital Chair of Sechenov University presents educational materials in Russian (https://dk.sechenov.ru/). Biomedical teaching includes the areas of telemedicine, e-Health, pharmaceutics (Lebedev et al. 2021; Koshechkin et al. 2022).

To overview regional educational initiatives in Russia we acknowledge the educational courses developing at I.Kant Baltic Federal University in Kaliningrad, St.-Petersburg Institute of Bioinformatics, Novosibirsk State University in Novosibirsk, Irkutsk State University in Irkutsk, Far Eastern Federal University in Vladivostok, Sirius Technological University in Sochi, and Peoples' Friendship University of Russia (RUDN University) in Moscow. Note NGS (next-gen sequencing) school for young scientists organized by Medical Genetics Centre of the Russian Academy of Sciences (RAS) (https://ngs.med-gen.ru/). The Institute of Cytology and Genetics of Siberian Branch of the RAS in Novosibirsk had supported educational Schools on bioinformatics for more than a decade (Baranova and Orlov, 2016).

As an example of the output of the courses, the application of gene network reconstruction and annotation have been effective for the education of medical students' leading to the students' publications (Karpyn et al, 2024; Savina et al., 2024).

## 6 KNOWLEDGE PRESENTATION IN LIFE SCIENCES FIELD

Biological knowledge presentation for the following referencing and education challenges AI approaches. Note work by (Kıyak et al., 2025) presenting descriptive study to evaluate the quality of KFQs (key-feature questions) generated by OpenAI's o3 model (Kıyak et al., 2025. The authors developed a reusable generic prompt for KFQ generation, designed in alignment with the Medical Council of Canada's KFQ development guidelines. Descriptive statistics were used to summarize checklist compliance and final acceptability ratings. Of the 20 KFQs, 3 were rated 'Accept as is' and 17 'Accept with minor revisions'; none required major revisions or were rejected. The overall compliance rate across checklist criteria was 93.7 percents with perfect scores in domains such as key feature definition, scenario plausibility, and alignment between questions and scenarios. Lower performance was observed for inclusion of genuinely harmful 'killer' responses (50 percents), plausibility of distractors (77.8), and active language use in phrasing the question (80 percents). The findings showed that an LLM, guided by a structured prompt, can generate KFQs that closely adhere to established quality standards, with most requiring only minor refinements (Kıyak et al., 2025).

Despite cross-referencing, most existing work and surveys remain fragmented, focusing on isolated tasks such as idea generation or experiment design without addressing how these components fit within the broader discovery process. To bridge this gap, the EXHYTE cycle, an iterative framework that formalizes scientific discovery as a sequence of Exploration, Hypothesis generation, and Testing was developed (ExHyTe) (Hasib et al., 2025).An accompanying website

with paper summaries and an LLM-powered interactive survey based on EXHYTE is available (https://webapps.crc.pitt.edu/exhyte/).

The accelerating growth of the biomedical literature makes it increasingly difficult to keep pace with connections between biological entities emerging across biomedical research. Recently developed automated means of generating hypotheses can generate many more hypotheses than can be easily tested. One such approach involves literature-based discovery (LBD) systems such as Serial Kinder-Miner (SKiM), which surfaces putative A-B-C links derived from term co-occurrence. LLMs have the potential to automate much of this curation step, but standalone LLMs are hampered by hallucinations, lack of transparency in information sources, and inability to reference data not included in the training corpus (Freeman et al., 2025).

SKiM-GPT, a retrieval-augmented generation (RAG) system that combines SKiM's co-occurrence search and retrieval with frontier LLMs to evaluate user-defined hypotheses was presented. For every chosen A-B-C SKiM hit, SKiM-GPT retrieves appropriate PubMed abstract texts, filters out irrelevant abstracts with a fine-tuned relevance model, and prompts an LLM to evaluate the user's hypothesis, given the relevant abstracts. SKiM-GPT is open-source (https://github.com/stewart-lab/skimgpt) and available through a web interface (https://skim.morgridge.org), enabling both wet-lab and computational researchers to systematically and efficiently evaluate biomedical hypotheses at scale (Freeman et al., 2025).

Objective Structured Clinical Examinations (OSCEs) are used as an evaluation method in medical education, but require significant pedagogical expertise and investment, especially in emerging fields like digital health. Large language models, such as ChatGPT (OpenAI), have shown potential in automating educational content generation. However, OSCE generation using LLMs remains underexplored (Zouakia et al., 2026) Structured prompting strategies, particularly agents' simulation, enhance the reliability and usability of LLM-generated OSCE content. These results support the use of artificial intelligence in medical education, while confirming the need for expert validation.

LLMs have shown remarkable capabilities in algorithm design, but their effectiveness in solving data science challenges in real-world settings remains poorly understood (Ma et al., 2025). A classroom experiment in which graduate students used LLMs to solve biomedical data science challenges on Kaggle, focusing on tabular data prediction revealed potential of LLMs to design competitive machine learning solutions, even when used by nonexperts.

## 7 CONCLUSIONS

Bioinformatics education in the forms of short courses, distant-learning, self-learning programs, and now generative models passed its way to modern development of AI tools. We conclude this review by mentioning new applications oof in Big Data and Machine Learning methods in education. Besides the research problem, AI has diverse effects on education, both positive and negative.

The perspectives on how to integrate tools like ChatGPT into bioinformatics have been discussed recently (Patel et al., 2024; Phan et al., 2024) including the bioinformatics education (Rigas et al., 2025). Generative artificial intelligence has delivered promising results in drug discovery and development (Gangwal et al., 2024).

The Open Educational Resources (https://en.unesco.org/themes/building-knowledge-societies/oer) supported by UNESCO also include AI tools for education courses online. The topics of health data science education, the training of biomedical students also rely to bioinformatics tools (Rohani et al., 2024; Sedlakova et al., 2025). The usage of ChatGPT in medical education among faculty and students had been discussed (Abouammoh et al., 2024). It can become an assisting tool for students as an additional exercise in informatics (Chan et al., 2025).

Note again grand challenges in bioinformatics related to AI - AI-enhanced lab automation, the Research-CoPilot, the convergence of omics and synthetic biology, and the integration of bioinformatics with spatial biology and 3D structure prediction for macromolecules (Khan et al., 2024). As the clinical adoption of deep learning algorithms, a subfield of AI, progresses, concerns have arisen regarding the impact of AI biases and discrimination on patient health (Ueda et al., 2024).

Integration of LLM (Large Language Models) into learning and teaching practice enhance digital competitiveness for medical and health science students (Jowsey et al., 2023). Latest problems relate to ethical concerns on AI (Resnik and Hosseini, 2025; Zouakia et al., 2026).

ACKNOWLEDGMENTS

The work was supported by Russian Science Foundation.

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
