# OpenReview forum: "Trends in IT and Life Sciences Education based on AI Tools"
_mathai.club/MathAI/2026/Conference — MathAI 2026 Conference Submission_

### Official Review · Reviewer_enfm · 2026-03-12
**This article offers a review of current trends involving large language models (LLMs), interactive learning platforms, and online courses in education.**

**Rating:** 4
**Confidence:** 4

**Review:**

The article's main advantage is the discussion of recent studies from 2025–2026 and a substantial list of references, which aligns with the review format. Nevertheless, for a conference focused on the 'Mathematics of Artificial Intelligence' (MathAI), the paper is deficient in mathematical depth, original methodologies, proprietary research, and detailed comparative evaluations.

---

### Official Review · Reviewer_cAhj · 2026-03-13
**A useful article. Weak acceptance. The paper reviews trends in the use of AI tools in IT and Life Sciences.**

**Rating:** 5
**Confidence:** 3

**Review:**

Summary: The paper reviews trends in the use of AI tools (specifically Large Language Models like ChatGPT) in IT and Life Sciences education. It surveys applications in medical decision support, bioinformatics training, active learning, gamification, and knowledge presentation. It highlights specific educational initiatives in Russia and discusses the integration of generative AI into curricula.

Strengths:
Relevant Topic: The use of AI in education is a timely and important subject.
Clear Structure: The paper is organized logically, making it easy to read.
Breadth of Coverage: It touches on many aspects of the educational ecosystem, from software tools to pedagogical strategies.

Weaknesses:
Lack of Mathematical Content: There is no mathematical rigor, no new models, and no theoretical analysis. It is a pure survey, which does not meet the criteria for a conference focused on the Mathematics of AI.
The paper summarizes the abstracts of other papers without providing a deep critical synthesis or a new theoretical framework.